# Pilot Study for Isolation of Stromal Vascular Fraction with Collagenase Using an Automated Processing System

**DOI:** 10.3390/ijms25137148

**Published:** 2024-06-28

**Authors:** Gershon Zinger, Yoav Gronovich, Adi Maisel Lotan, Racheli Sharon-Gabbay

**Affiliations:** 1Department of Orthopedic Surgery, Hand Unit, The Eisenberg R&D Authority, Shaare Zedek Medical Center, Faculty of Medicine, Hebrew University of Jerusalem, Jerusalem 9103102, Israel; rachelish@szmc.org.il; 2Department of Plastic & Reconstructive Surgery, Shaare Zedek Medical Center, Faculty of Medicine, Hebrew University of Jerusalem, Jerusalem 9103102, Israel; yoavg@szmc.org.il (Y.G.); lotanadi@szmc.org.il (A.M.L.)

**Keywords:** stromal vascular fraction, SVF, adipose tissue, regenerative cells, cell count, ASCs, ACS

## Abstract

There are many potential therapeutic applications for autologous adipose-derived stromal cells. These cells are found in a heterogeneous population isolated from adipose tissue called the stromal vascular fraction (SVF). Closed automated systems are available to release cells from the adherent stroma. Here, we test one system to evaluate the heterogeneous output for yield, purity, cellular characterization, and stemness criteria. The SVF was isolated from three donors using the Automated Cell Station (ACS) from BSL Co., Ltd., Busan, Republic of Korea. The SVF cellular output was characterized for cell yield and viability, immunophenotyping analysis, pluripotent differentiation potential, adhesion to plastic, and colony-forming units. Additionally, the SVF was tested for endotoxin and collagenase residuals. The SVF yield from the ACS system was an average volume of 7.9 ± 0.5 mL containing an average of 19 × 10^6^ nucleated cells with 85 ± 12% viability. Flow cytometry identified a variety of cells, including ASCs (23%), macrophages (24%), endothelial cells (5%), pericytes (4%), and transitional cells (0.5%). The final concentrated product contained cells capable of differentiating into adipogenic, chondrogenic, and osteogenic phenotypes. Furthermore, tests for SVF sterility and purity showed no evidence of endotoxin or collagenase residuals. The ACS system can efficiently process cells from adipose tissue within the timeframe of a single surgical procedure. The cellular characterization indicated that this system can yield a sterile and concentrated SVF output, providing a valuable source of ASCs within the heterogeneous cell population.

## 1. Introduction

Recent studies have highlighted the potential of mesenchymal stromal cell (MSC) therapy as an important element in improving biological homeostasis within the human body [1]. MSCs can be harvested from various sources, including placental tissue, bone marrow, and subcutaneous fat [2]. Among these sources, adipose tissue is exceptional for both its abundance and ease of harvesting. Contained in the adipose tissue is a high concentration of mesenchymal stromal/stem cells (MSCs) [3]. The liposuction procedure involves a minimally invasive procedure under local anesthetic. Donor sites for liposuction typically include the abdomen, flanks, and thighs. Lipoaspiration is performed under local anesthetic, is associated with minimal discomfort and has a long history of safety [4].

Studies in the field of regenerative medicine have demonstrated the versatile application of autologous ASCs in various treatments, both local and systemic [5,6,7,8,9]. ASCs have the potential to transform into different cell types, including adipocytes, osteoblasts, and chondrocytes [10], facilitating tissue repair and regeneration [11]. Both preclinical and clinical investigations have consistently demonstrated ASC therapeutic effects on local tissue, including antiapoptotic, anti-inflammatory, and angiogenic tissues [12,13]. The primary modes of action of ASCs appear to be immunomodulatory and via paracrine signaling [14].

In plastic surgery, SVF has been used to improve wound healing [10,11] including tissue regeneration and scar reduction [8]. The orthopedic applications for SVF include the treatment of osteoarthritis via its anti-inflammatory effects and via cartilage repair [9]. SVF injections for arthritis can provide an option for those patients when traditional conservative therapies, such as physical therapy and steroid injections, have failed, but where they are not ready for joint replacement [15].

The processing of lipoaspirate to extract and concentrate cellular components can be achieved through two primary methods: enzymatic and mechanical. The enzymatic method involves the use of enzymes, primarily collagenase, to break down the extracellular matrix of adipose tissue. Following enzymatic digestion, the mixture undergoes centrifugation and/or filtration to remove the oil, fluid, and debris, leaving behind the cellular components. This resulting heterogeneous mixture is known as the stromal vascular fraction (SVF). Conversely, the mechanical method relies on mechanical forces to disrupt adipose tissue. Mechanical methods may include homogenization, centrifugation, and filtering. Subsequently, the oil and fluid are discarded, leaving behind more concentrated cellular elements. Mechanical systems offer advantages such as reduced cost and time savings. However, enzymatic processing has been shown to be more efficient at isolating and concentrating the desired nucleated and regenerative cells. In some reports, enzymatic methods yield more than 10 times the concentration compared to mechanical methods [16]. Yet, enzymatic methods have disadvantages, including equipment cost, incubation time, and regulatory restrictions. 

In recent years, automated closed systems have emerged for processing lipoaspirate to isolate the SVF. These automated systems offer numerous advantages, including enhanced efficiency, increased reproducibility, reduced dependence on highly trained personnel, and improved point-of-care treatment by enhancing the sterility of SVF production. Various commercial systems are available on the market that are either semi-automated or fully automated. For example, the Sepax^®^ system employs an automated centrifugation and cell separation system to isolate SVF [17]. The Cha-Station, developed by CHA Biotech in Kangnamgu, Republic of Korea, is a closed semi-automated processing system [18]. Additionally, the Cytori Celution^®^ (Cytori Therapeutics, Inc., San Diego, CA, USA) utilizes automated cell separation with sterile handling to obtain high-quality SVF [19,20]. These systems integrate automated or semi-automated cell separation in conjunction with enzymatic digestion. In 2013, Aronowitz conducted a comparison of four commercially available SVF isolation systems. The study involved SVF characterization, an investigation of residual collagenase activity, and an assessment of processing economics [18].

In this study, we evaluated the performance of the Automated Cell Station (ACS, BSL Co., Ltd. in Republic of Korea) device using three volunteers. This system streamlines the production of SVF from adipose tissue using a device with two separate compartments: an incubator and a computerized centrifuge. The system employs sterile single-use kits and an automated cell processing unit (ACPU) to process the digested lipoaspirate before its placement into the pre-programmed centrifuge system that washes and concentrates the SVF.

Despite the widespread clinical use of the ACS product [Cell Surgical Network^®^, Mountain View, CA, USA], there is a lack of scientific publications assessing the nature of the end product, the SVF. The International Federation for Adipose Therapeutics and Science (IFATS) and the International Society for Cellular Therapy (ISCT) proposed three minimal criteria for the definition of ASCs: (1) plastic adherence; (2) specific surface antigen expression of CD73, CD90, and CD105 and a lack of expression of CD11b, CD14, CD19, CD45, and HLA-DR; and (3) trilineage differentiation potential into adipocytes, chondrocytes, and osteoblasts [21,22]. In this study, SVF was evaluated to determine if it meets the minimal criteria for stemness. In addition, the SVF cellular content was characterized and additional analysis was conducted to confirm that the SVF met additional regulatory requirements.

## 2. Results and Discussion

Lipoaspirate was collected from the flanks of three volunteers during three different time sessions. The average amount of lipoaspirate collected was 82 ± 5.3 mL (Figure 1A). After decantation, the remaining fat tissue was 52 ± 6.9 mL (Figure 1B). After 60 min of incubation the ACPU was filled with the digested fat mixture (Figure 1C) and placed into the ACS centrifuge (the upper part of the ACS device, Figure 1D). The centrifuge is programmed for 3 cycle of washing and centrifugation (lasting 23 min). The final SVF product is collected from the ACPU with an average volume of 7.9 ± 0.5 mL (Figure 1E). The average nucleated cell concentration was 2.40 ± 1.9 × 10^6^ nucleated cells/mL (total of 19 × 10^6^ nucleated cells), with 85 ± 12% viability. Relative to the initial lipoaspirate volume, the nucleated yield was an average of 2.3 × 10^5^ cells per ml of lipoaspirate. All three samples were negative for endotoxin (<0.01 EU/mL) and negative for collagenase. The collagenase activity test found no residual levels of collagenase (Figure 2). 

### 2.1. Cell Culture

Adherent cells of SVF were observed at 24 h after seeding in culture. After 3 weeks of culturing, the adherent cells coalesced into colonies (Figure 3A,B). The average number of colonies was 75 ± 13 colonies/cm^2^ (Figure 3C). 

### 2.2. Analysis of Multipotency of SVF Product

The multilineage differentiation potential of the SVF obtained from the ACS device was evaluated by confirming the ability of adherent cells obtained in the SVF to differentiate to adipocytes, chondrocytes, and osteoblasts. Adherent cells obtained from the SVF differentiated into all three mesodermal lineages (Figure 4 and Figure 5, upper panels). The cultured SVF that was treated with differentiation media and stained with Oil Red O, Alcian Blue, and Alizarin Red S (ARS) were positive for adipogenic, chondrogenic, and osteogenic differentiation, respectively. In addition, low positive staining was also detected for the control cells that were treated with MSC maintenance medium (Figure 4 and Figure 5, lower panels), indicating the strong multipotency of the ASCs that reside in the SVF.

### 2.3. Flow Cytometry Analysis

To determine the cellular composition of SVF, flow cytometry analysis was performed to identify the different cell populations based on their surface expression markers. After RBC lysis, most of the characterized cells were either positive for CD45 or CD34, which are markers for WBC and stem cells, respectively. To assess the cell identity and frequency, a combination of CD31 (endothelial marker) and CD146 (perivascular marker) was tested. Figure 6A shows a representative FACS image of separated subsets within the SVF. The average percentage of ASCs characterized as CD34+ and CD146− was 23%. Of those, 95% were positive for CD90, 58% for CD105, and 94% for CD73 (Figure 6B). The average percentage of macrophages characterized as CD206+ was 24%; the average percentage of endothelial cells characterized as CD34+, CD31+, and CD146+ was 5%; the average percentage of pericytes characterized as CD34−, CD31−, and CD146+ was 4%; and the average percentage of transitional cells characterized as CD34+, CD31−, and CD146+ made up only a small fraction of 0.5% (Figure 7). The viability of cells was verified by 7-AAD staining and averaged 85 ± 12%.

### 2.4. Media Fill Process

No signs of contamination or growth of microorganisms were found after 14 days of incubation. The simulation results showed that the media fill process was effective in maintaining sterility and preventing microbial contamination. Therefore, this simulation provides important information for ensuring safety and efficacy in the production of sterile products.

## 3. Materials and Methods

### 3.1. Liposuction Procedure

Three male participants, with a mean age of 36 ± 16 years and an average body mass index of 28.2 ± 3.4 kg/m^2^, were entered into the study after they provided informed consent by signing an IRB-approved consent form (Helsinki 0497-20-SZMC). Liposuction was performed after an initial infusion of local anesthesia using lidocaine 0.5% with epinephrine buffered with bicarbonate. With the patient prone, the anesthetic was infused into each flank using a 16-gauge 6-inch blunt-tip cannula with 4 small side ports. The cannula was first used to break up the fatty tissue through 5–6 passes in the subcutaneous tissue. The lidocaine solution was administered to the flanks in a triangular pattern. After waiting 30 min, the manual liposuction procedure was performed using a cannula with a 3 mm outer diameter and a length of 12 mm with multiple side ports connected to a 10 mL syringe. This was divided into two sterile 60 mL syringes that were left in a vertical position for decantation. The liquid waste at the bottom of the syringes was discarded. 

### 3.2. Stromal Vascular Fraction (SVF) Sterile Isolation

The lipoaspirate was decanted in the two 60 mL syringes for approximately 3 min. The adipose tissue was washed three times with an equal amount of lactated ringer (LR) solution. The washed fat was passed through a 0.6 mm Adinizer filter (BSL Co., Ltd., Republic of Korea, according to the BCL protocol, this is not a required step) prior to the addition of collagenase (Liberas MNP-S GMP grade, Roche, Heidelberg, Germany) containing 11 Wunch units. The mixture was incubated for 60 min in the ACS incubator (BSL Co., Ltd., Republic of Korea) at 37 °C. The digested fat was passed through a 0.1 mm Adinizer filter (according to the BCL protocol, this step would be with a 0.6 mm Adinizer filter) and injected into the ACPU single-use device. The washing solution chamber in the ACPU was filled with 150 mL of LR solution. The unit was placed inside the ACS centrifuge, and the SVF program was initiated. After 23 min, the final ACPU was removed from the centrifuge and the SVF product was collected from the ACPU using a long-collection blunt needle. The final SVF was analyzed immediately in the operating room for cell count and cell viability, then transferred to the laboratory for additional SVF characterization.

### 3.3. Cell Count and Cell Viability

The nucleated cell count was measured using an automated cell counter, CellDrop^TM^ FL (DeNovix, Wilmington, DE, USA). The SVF sample was mixed with AO/PI (1:1) prior to direct pipette loading; then, an automated counter was used.

### 3.4. Endotoxin

The glucan concentration was measured by endotoxin testing. This included a spectrophotometer that utilizes disposable cartridges, i.e., Endosafe^®^ nexgen-PTS™ cartridges purchased from Charles River’s. The supernatant of the SVF sample was filtered through a 70 μm filter and centrifuged again at 1500× *g* for 5 min. The supernatant was filtered and diluted to 1:100 using LAL reagent. According to the manufacturer’s instructions, 25 μL of the diluted sample was inserted into each of the four sample reservoirs of the cartridge. The optical density of the wells was measured using the Endosafe^®^-PTS™ and analyzed against an internally archived standard curve. By design, the cartridge technology automatically performed a duplicate sample and duplicate positive control LAL test, thereby satisfying the harmonized USP/EP bacterial endotoxin test (BET) for LAL testing. 

### 3.5. Collagenase Residual

The detection of collagenase residual in the three SVF samples was carried out using the Collagenase Activity test kit for in-process control (Cat. No. 08074461001, Roche Diagnostics, Rotkreuz, Switzerland). The kit included the assessment of a five-point calibration curve for the enzymatic activity quantification of Collagenase I and Collagenase II in Liberase blends. The hydrolysis kinetics of the chromogenic substrate FALGPA, a labeled tetrapeptide resembling the primary structure of collagen, was measured with Liberase standard; the three SVF samples and collagenase-free blend were used as a negative control. Using the multiwall plate reader, the absorbances at 340 nm of 15 data points (1 measure per minute) were recorded. The maximal reaction velocities (Vmax (ΔAbs/min), linear slope) and enzymatic activity concentration were determined. 

### 3.6. Culture of SVF

Freshly isolated SVF (2.5 mL) was centrifuged at 400× *g* for 5 min. The pellet of cells was re-suspended with MSC NutriStem^®^ XF complete medium (Catalog No: 05-200-1A, Sartorius, Göttingen, Germany). The cells were seeded in three concentrations (10, 5, and 2.5 × 10^6^ cell/mL) into a pre-coated (MSC Attachment Solution, diluted 1:100 in DPBS, Cat. No. 05-752-1, Sartorius, Germany) T25-Flask, pre-coated 96-well F plates (adipogenic and osteogenic differentiation), and a non-coated 96-well U bottom plate used for chondrogenic differentiation. The cells were incubated at 37 °C and 5% CO_2_. After 24 h, the seeding medium was changed with fresh MSC NutriStem^®^ XF complete medium composed of serum-free (SF) and xeno-free (XF) medium, with all components defined and of non-xenogenic origin, including proteins. The medium was replaced every 3 days. After 7 days, the differentiating assay was carried out. 

### 3.7. Cell Differentiation

The ability of cells to differentiate into mesodermal lineages was tested in adipogenic, chondrogenic, and osteogenic differentiation media: MSCgo™ Adipogenic Differentiation, a serum-free and xeno-free medium (05-330-1B) containing MSCgo™ Adipogenic XF Supplement Mix I (05-331-1-01); MSCgo™ Adipogenic XF Supplement Mix II (05-332-1-15); MSCgo™ Chondrogenic Differentiation Medium basal medium (05-220-1B), containing MSCgo™ Chondrogenic XF Supplement Mix (05-221-1D); and MSCgo™ Osteogenic Differentiation Media (05-220-1B). The media and supplements were purchased from Sartorius, Germany. When 80% confluence was reached, after 1–2 weeks of culture in MSC NutriStem^®^ XF complete medium, the medium was changed to a differentiation medium. The cells were incubated for a total of 3–4 weeks prior to staining. The adipogenic maintenance schedule included one cycle of adipogenic differentiation for 6–8 days, with the MSCgo™ Adipogenic complete medium being changed every 3–4 days. After the adipogenic induction, the medium was replaced with a maintenance medium, MSC NutriStem^®^ XF, for a period of 3–4 days. Cells were stained with Oil Red O (Sigma, Livonia, MI, USA) for the detection of lipid droplets accumulating intracellularly, which are an indication of mature adipocytes. Chondrogenesis was confirmed by using Alcian Blue (Sigma) to stain proteoglycan aggrecan, a component of cartilage. The osteogenesis evaluation was performed using Alizarin Red S (ARS) for culture mineralization staining. 

### 3.8. Flow Analysis

The immunophenotyping of freshly isolated SVF was performed by flow cytometry using a BD Canto II cytometer with 3 lasers (blue (488 nm), red (633 nm), and violet (405 nm), BD, Franklin Lakes, NJ, USA). The removal of erythrocytes was carried out using RBC lysis buffer (BD FACS lysing solution, 349202, USA). The following monoclonal antibodies (MAbs), conjugated to various fluorochromes, were used in single-cell suspensions of SVF to differentiate between the different cell populations: CD34, CD31, CD106, CD206, CD146, CD73, CD90, CD45, CD105, CD235a, CD44, CD34, CD29, CD326, and HLA-DR (antibodies were purchased from BD Biosciences). To assess the SVF cell count and viability, TrueCount beads and 7-amino-actinomycin (7-AAD) were used (BD Biosciences, Franklin Lakes, NJ, USA).

### 3.9. CFU

SVF cells (1.78 × 10^5^) were seeded in 3 wells of a 96-well plate (0.33 cm^2^) in MSC NutriStem^®^ XF complete medium. After 24 h, the medium was changed and the non-attached cells were removed. Cells were incubated in a 5% CO_2_ incubator at 37 °C and 95% humidity for 3 weeks. Stained (Alizarin Red S) colonies were counted. The results were expressed in colonies per cm^2^.

### 3.10. Media Fill Process

A simulation of the media fill process using Tryptic Soy Broth (TSB) was conducted to evaluate the efficacy of the liposuction procedure and SVF isolation in terms of maintaining sterility. The TSB was processed in the same manner that the fat was processed, as described above. Specifically, after the first “incubation” in the ACS, the TSB was placed into an ACPU kit and processed using an automated centrifuge, similarly to the procedure for SVF production described above. The resulting “SVF” was then incubated for a period of 14 days and assessed for any signs of contamination or growth of microorganisms. Three separate media fill processes were conducted.

## 4. Conclusions

Adipose tissue has gained significant attention as a source for regenerative treatments. Processed lipoaspirate, yielding SVF produced using enzymes or mechanical methods, has shown clinically proven to have therapeutic potential in different medical applications. In recent years, automated systems have been developed for processing lipoaspirate. They have the advantages of enhanced efficiency, increased reproducibility, reduced dependence on highly trained personnel, and improved sterility of the SVF end product. To fully harness ASCs’ capabilities and understand the cellular properties of SVF, its characterization is essential. This will lead to more effective treatment strategies and enhanced medical outcomes.

The results of lipoaspirate processing using automated equipment lack unbiased studies examining the quantity and quality of the resultant SVF. The ACS system tested in this study is already being used clinically. The findings here focus on the cellular characterization of the ACS system’s SVF end product. It is crucial to note that no compensation or complimentary products were received, and the ACS company had no involvement in either the design or the outcomes of this preliminary study, which serves as a precursor to a larger project. The importance of this research lies in its potential clinical applications. 

The analysis indicated that the ACS system successfully produced SVF high in quantity and quality. The final SVF product had an average nucleated cell count of 2.40 ± 1.9 × 10^6^ nucleated cells/mL (total of 19 × 10^6^ cells in average volume of 7.9 ± 0.5 mL final product) with 85% viability. In addition, 23% of the nucleated cells were ASCs. These results are similar to those found by Aronowitz et al. in their comparison of four other automated cell separation systems [23]. In that comparison, the Cytori StemSource 900/MB system yielded 1.01 × 10^5^ nucleated cells per ml of fat, with 10.5% being ASCs, compared to the ACS device evaluated here with 3.65 × 10^5^ nucleated cells per ml of decanted fat.

According to the criteria proposed by the Mesenchymal and Tissue Stem Cell Committee of the ISCT, human mesenchymal stem cells are defined by their capacity to adhere to plastic; to create colonies; to express specific antigens; and to differentiate into adipocytes, chondrocytes, and osteoblasts [21]. Here, all of the above criteria were confirmed for each of the three SVF samples, confirming the mesenchymal stem cell potential of the SVF produced by the ACS. 

The therapeutic potential of paracrine activity from isolated SVF cells is important. However, in this study, the SVF proteome was not analyzed. Enzymatic production for characterizing these factors may not be ideal due to several reasons. The cell isolation process involves multiple wash steps, potentially diluting or depleting the paracrine activity. In clinical applications, fresh SVF is applied immediately post isolation. It is likely that, after deployment, the stromal cells will produce a significant paracrine activity. However, this study is focused on the analysis of fresh SVF to show that the material contains MSCs. Utilizing cell culture models, even for a minimum time of two days, as carried out by El-Habta [24], enables the analysis of SVF secretion in response to biomimetic conditions. This type of analysis after culture may offer more insightful data on the therapeutic mechanisms.

In addition to these criteria, important point-of-care tests were conducted that confirmed that no contamination or endotoxin could be found. Additionally, the absence of residual collagenase, as confirmed by the collagenase activity test, further supports the efficiency of the ACS device in producing a purified SVF product.

Additional important factors considered in an automated processing system include the final volume of SVF and the disposable costs. The final volume of SVF produced by the Cytori StemSource 900/MB system is 5 mL [23]. Clinically, the volume can be increased with the addition of saline, but this will dilute the nucleated cell concentration. In cases where a larger volume is required, the ACS device is preferred, offering 8–10 mL. The disposable cost of the Cytori StemSource 900/MB system is USD 2400 per 120–360 mL of processed fat [23], whereas the ACS device costs USD 500 per 50 mL. 

The ACS automated system presents an elegant solution for processing lipoaspirate and generating high-quality SVF for therapeutic applications. By automating the steps of washing and centrifugation within a closed-loop system (ACPU), the ACS device minimizes the potential for human error and contamination, resulting in consistent and reliable SVF production. Furthermore, as the procedure is conducted within the operating room, the fat is processed under controlled air conditions and stringent cleanliness standards. This ensures the maintenance of sterility and minimizes the risk of contamination throughout the processing steps.

However, there are several disadvantages associated with the ACS system that should be considered. One significant drawback is the upfront cost of the device, which typically costs USD 50,000. This initial investment may pose financial challenges for smaller clinics or healthcare facilities with limited budgets. Additionally, the single-use unit (ACPU) cost is estimated at USD 500 per unit. Finally, the cost of GMP standard enzymatic collagenase required for SVF production and redeployment adds an additional estimated cost of approximately USD 475 per 5 mg.

In conclusion, this pilot study characterized the end product of an automated lipoaspirate processing system for the isolation of SVF. The automated system efficiently produced a high concentration of viable nucleated cells that exhibited human mesenchymal stem cell capabilities. In considering the use of the ACS system in clinical practice, healthcare providers must carefully weigh up the advantages and disadvantages of an automated system to ensure the cost-effective and sustainable delivery of SVF-based therapies to patients in need. Further research and larger-scale trials are warranted in order to fully establish the clinical applicability of the ACS device for SVF isolation and its therapeutic potential for various medical conditions.

## Figures and Tables

**Figure 1 ijms-25-07148-f001:**
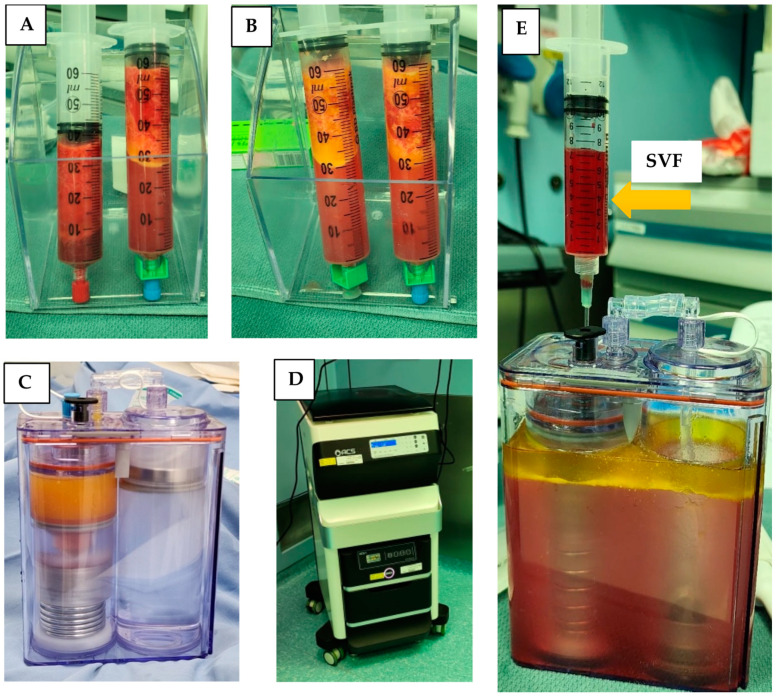
Isolation of SVF using the ACS machine in the OR. (**A**) Decantation of lipoaspirate; (**B**) decantation after washing lipoaspirate; (**C**) ACPU filled with the digested fat (left side) and washing solution (right side); (**D**) ACS device with the incubator below and automated centrifuge above; (**E**) ACPU after processing with SVF being collected in the syringe and waste material remaining in the container.

**Figure 2 ijms-25-07148-f002:**
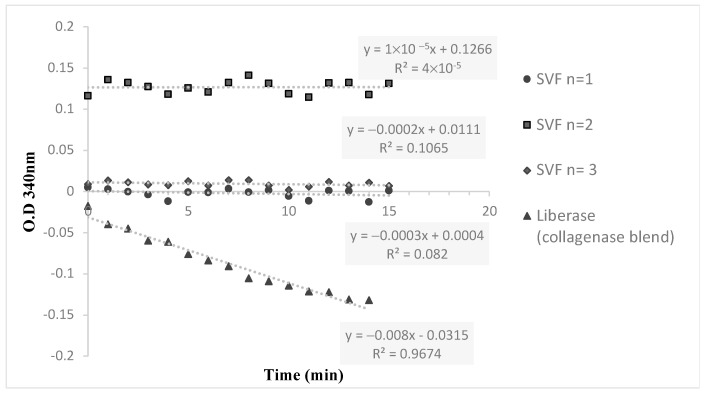
The hydrolysis kinetics of collagenase substrate (FALGPA) for 3 different SVF samples and a positive control (Liberase).

**Figure 3 ijms-25-07148-f003:**
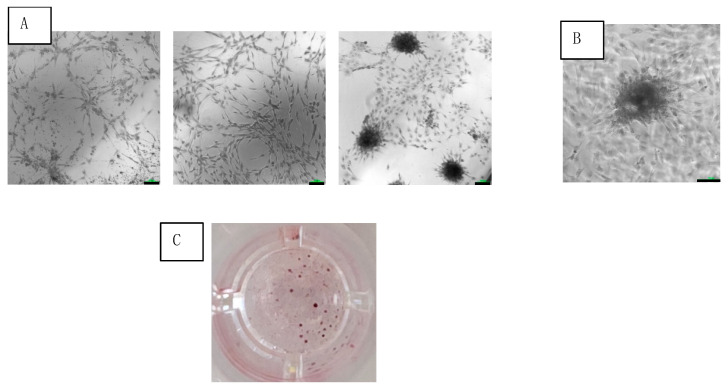
Colony formation of SVF cells cultured under osteogenic medium at 3 weeks. (**A**) Increasing cell concentrations (from left to right) seen under light microscopy with 10× magnification (scale bar = 100 μm). (**B**) Colony magnification at 20× (scale bar = 100 μm). (**C**) Photo of single well out of 96-well plate of stained colonies.

**Figure 4 ijms-25-07148-f004:**
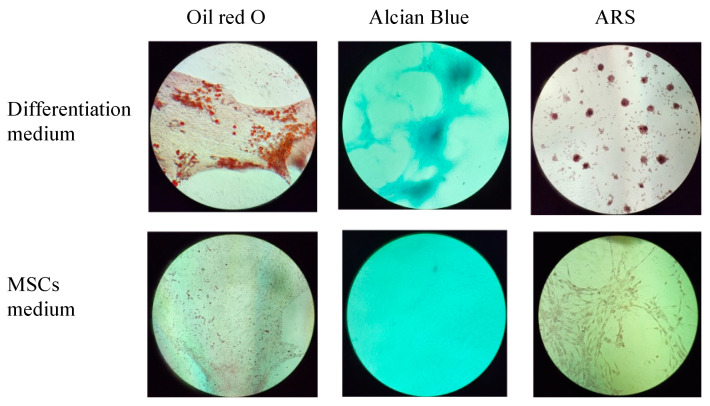
Morphological analysis of in vitro ASCs monolayer differentiation (**upper panel**) under adipogenic, chondrogenic, and osteogenic media conditions and control; (**lower panel**) ASCs under MSCs maintenance medium after 16–21 days, followed by Oil Red-O, Alcian Blue, and ARS staining and seen under light microscopy with 10× magnification.

**Figure 5 ijms-25-07148-f005:**
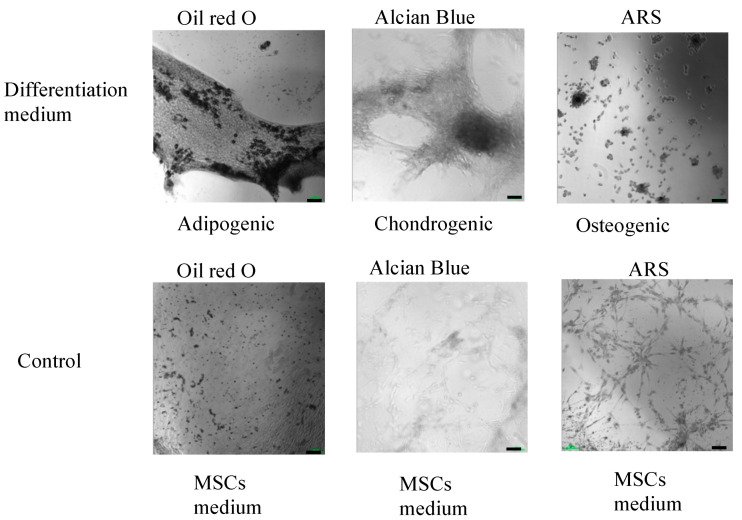
High-resolution, black and white morphological analysis of in vitro ASCs monolayer differentiation (**upper panel**) under adipogenic, chondrogenic, and osteogenic media conditions and control; (**lower panel**) ASCs under MSCs maintenance medium after 16–21 days, followed by Oil Red-O, Alcian Blue, and ARS staining and seen under light microscopy with 10× magnification (scale bar = 100 μm).

**Figure 6 ijms-25-07148-f006:**
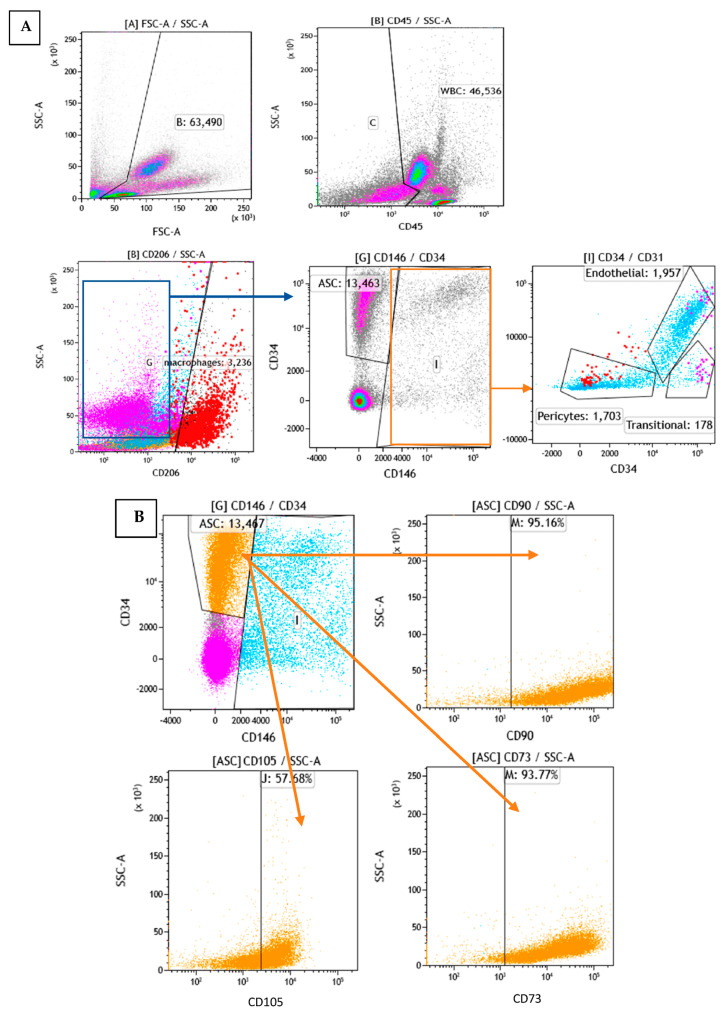
Flow cytometry analysis of SVF: (**A**): Analysis of subpopulations by phenotype (different colors represent different subpopulations). (**B**): Analysis of the orange subpopulation with CD90, CD105, and CD73 markers.

**Figure 7 ijms-25-07148-f007:**
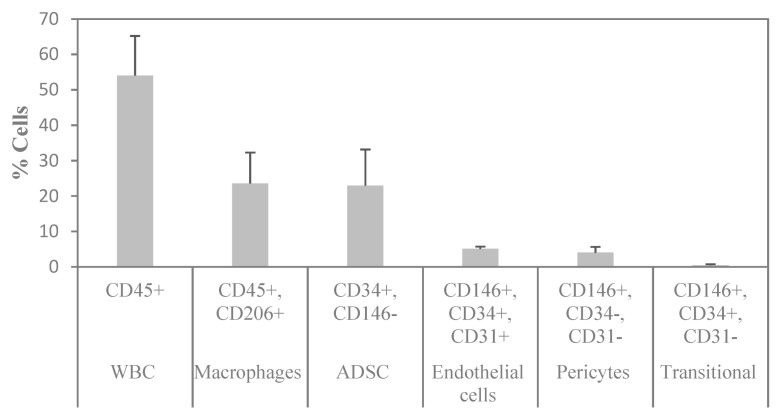
Summary of the flow cytometry analysis of SVF. n = 3 (mean ± SD).

## Data Availability

Data from this study are available on request at gershonzinger@szmc.org.il.

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
