# Peer review of "Pilot Study for Isolation of Stromal Vascular Fraction with Collagenase Using an Automated Processing System"

_ijms, 2024, doi:10.3390/ijms25137148_

Round 1
Reviewer 1 Report (New Reviewer)
Comments and Suggestions for Authors
The biggest disadvantage of the proposed manuscript is the lack of information for using magnification during the microscopic observations. The magnitude is very different but in the text below it is explained as 20x. For example, in Figure 3, the size of the cells involved in colonies is bigger on day 3, after that they become smaller on day 6 and again bigger on day 9.
The other general remark is related to the multilineage differentiation potential. Concerning the adipogenic differentiation it is not clear if the picture is taken from a single cell. Again, what is the magnitude of applied magnification? It would be better if the larger field of differentiated cells is shown. There is no well-visible monolayer. The quality of the pictures is not appropriate, especially for Figure 3.
ADSCs should be replaced with ASCs in the whole manuscript according to the ISATS, 2008
Page 1, lines 78-82: Could you explain the abbreviation of “SVS”?

Author Response
|
Reviewer
|
comments |
Answer |
Change |
|
Reviewer 1
|
The biggest disadvantage of the proposed manuscript is the lack of information for using magnification during the microscopic observations. The magnitude is very different but in the text below it is explained as 20x. For example, in Figure 3, the size of the cells involved in colonies is bigger on day 3, after that they become smaller on day 6 and again bigger on day 9.
|
To address this concern, we have revised figure 3. Instead of displaying the colonies formation as a factor of time, we present colony formation as a factor of concentration. Figure 3 displays different dilutions all after 3 weeks of culture.
Thank you for bringing this to our attention. |
Line 152 figure 3. Lines 153-157 figure 3 legend. |
|
Reviewer 1
|
The other general remark is related to the multilineage differentiation potential. Concerning the adipogenic differentiation it is not clear if the picture is taken from a single cell. Again, what is the magnitude of applied magnification? It would be better if the larger field of differentiated cells is shown. There is no well-visible monolayer. The quality of the pictures is not appropriate, especially for Figure 3.
|
Figure 4b that includes high resolution black and white microphotographs was added. Figure 4a was left to demonstrate color photomicrographs demonstrating the different cell types based on their staining. |
Lines 172-175 figure legend was rewrite. Figure 4b.was added. |
|
Reviewer 1
|
ADSCs should be replaced with ASCs in the whole manuscript according to the ISATS, 2008 |
ADSCs was replaced with ASCs throughout the manuscript |
|
|
Reviewer 1
|
Page 1, lines 78-82: Could you explain the abbreviation of “SVS”? |
SVS is a typo – corrected to SVF (stromal vascular fraction introduced the abbreviation line 74) |
|

Reviewer 2 Report (New Reviewer)
Comments and Suggestions for Authors
Dear colleagues!
Overall, the study is of sound design used for descriptive assessment of and original system to obtain SVF from human adipose tissue. Microscope, FACS and differentiation assays are thorough and provide sufficient data for conclusions drawn.
Eventually, one may have the following comments:
1) microphotoraphs in "round-shaped" windows are ok yet they lack scale-bars
2) in page 15 figure legend is missing under the SVF station images
3) as of crucial points one may question the paracrine activity of isolated cells in obtained SVF. Secretome assessment has been made for SVF preparations and samples previously (https://www.ncbi.nlm.nih.gov/pmc/articles/PMC9762598/
https://www.researchgate.net/publication/349805250_Anti-apoptotic_effect_of_adipose_tissue-derived_stromal_vascular_fraction_in_denervated_rat_muscle
https://www.ncbi.nlm.nih.gov/pmc/articles/PMC6302486/) and it is well-established that pro-regenerative capacity of ADSC and some other cell types used in therapy is secretome-mediated. Thus, one may question why it has not been assessed in present work to support the efficacy of isolation procedure.
Regards, Reviewer
Comments on the Quality of English LanguageProviding and "review tracking" version of the manuscript makes it hard to read so I suggest to upload 2 versions of the paper to make it easier for the Reviewers
Author Response
|
Reviewer 2 |
1) microphotographs in "round-shaped" windows are ok yet they lack scale-bars
|
See above |
|
|
Reviewer 2 |
2) in page 15 figure legend is missing under the SVF station images
|
We were not able to find page 15 figure legend in the manuscript |
|
|
Reviewer 2 |
3) as of crucial points one may question the paracrine activity of isolated cells in obtained SVF. Secretome assessment has been made for SVF preparations and samples previously (https://www.ncbi.nlm.nih.gov/pmc/articles/PMC9762598/ https://www.researchgate.net/publication/349805250_Anti-apoptotic_effect_of_adipose_tissue-derived_stromal_vascular_fraction_in_denervated_rat_muscle https://www.ncbi.nlm.nih.gov/pmc/articles/PMC6302486/) and it is well-established that pro-regenerative capacity of ADSC and some other cell types used in therapy is secretome-mediated. Thus, one may question why it has not been assessed in present work to support the efficacy of isolation procedure.
|
This is a valuable point by the reviewer: We acknowledge the therapeutic potential paracrine activity of isolated SVF cells. We plan to optimize SVF isolation for therapeutic efficacy in a future research proposal that was already sent for grant approval. However, fresh SVF by enzymatic method may not be ideal due to several reasons. Enzymatic cell isolation processing involves multiple wash steps potentially dilute or deplete initially present paracrine activity. In clinical application, fresh SVF is applied immediately post-isolation. It is likely that after deployment, the stromal cells will produce a significant paracrine activity. However, this study is focused on the analysis of fresh SVF to show that the material contains MSCs. Utilizing cell culture models, even for a minimum time of two days, as done by El-Habta, enable the analysis of SVF secretion in response to biomimetic conditions. This type of analysis after culture may offer more insightful data on therapeutic mechanisms. |
Added to lines 372-381 |

This manuscript is a resubmission of an earlier submission. The following is a list of the peer review reports and author responses from that submission.
Round 1
Reviewer 1 Report
Comments and Suggestions for Authors
The authors present a pilot study from three human samples to show that the ACS device from Equipforskin successfully produces SVF with both good quantity and quality. Here my comments on this work.
There is now a great interest in the isolation of stromal stem cells for regenerative medicine. Different strategies (enzymatic, mechanical, or a combination of enzymatic and mechanical approaches) can be selected with different advantages and limitations. I think that from this work it is not clear why a researcher or a medical doctor should select this specific ACS device that is very expensive. The device is commercially available, then the presented data should help to understand the effectiveness of this method compared to other approaches. I think that this is not clear and more importantly 3 samples are not sufficient.
Regarding the Introduction section, the authors must discuss in more details the state of art. There are different strategies for isolation of the vascular stromal fraction: enzymatic, mechanical and combination of both approaches. The enzymatic method has some disadvantages in reproducibility, presence of contaminants and cost. Recently some devices were developed to avoid the use of the enzymatic treatment. Discuss this point and cite the paper from Semenzato et al.2023 Biomedicines PMID: 37189624. Moreover, at Lines 419-420 the authors wrote “in the context of lipoaspirate processing using automated equipment, there is insufficient objective information about the quantity and quality of the SVF produced”: this is not true; see for example the work by Semenzato et al., 2023.
In the Introduction the authors describe in details the procedures. They must simplify and use this section to introduce the state of art and to discuss in this context the presented findings.
Line 76-78 In this pilot work we characterized and analyzed the SVF that was isolated using the Automated Cell Separation (ACS) device. This analysis was performed in anticipation of a larger project. We received no compensation or free product, and the company was not involved in the design or results of this preliminary work. Move this part out from the introduction.
Figure 1 is at the end in the Materials and Methods section. Figure 1 must be in the results section and the procedure must be described in the proper way. The authors must organize better the manuscript and the relative findings.
Line 357 and Line 366: why the authors must specify the figure relative to the method described?
Lines 394-398 The authors must describe the staining procedures. In general, many procedures must be described, including for example flow cytometry.
The authors must show the nature of 3-D structures spheroids by staining with adipocyte and chondrocyte specific markers.
Perform a quantitative real time PCR to check the expression levels of adipocyte, chondrocyte and osteoblast specific markers to show that each SVF isolated samples is able to multilineage differentiate. Represents with dots-box plot.
Figure 7 Use a table to show the data of SVF composition from each human samples. Moreover, represents the data in the histogram as dots-box plot to see the data from each individual.
In summary the authors must deeply revise the presented manuscript.
Comments on the Quality of English LanguageThe manuscript needs revision for language and grammar.
Reviewer 2 Report
Comments and Suggestions for Authors
The authors describe a new automatic system that allows the extraction of stem cells from adipose tissue using collagenase.
The manuscript is among several methods currently being developed for the isolation of adipose stem cells, although current legislation in many countries prohibits the use of collagenase for clinical purposes.
In this regard, in the introduction the authors should cite the works of Raposio E and De Francesco F regarding the use of mechanical systems for the isolation of adipose tissue.
Furthermore, the correct terminology to indicate adipose stem cells is universally recognized as "ASC" and not ADSC, therefore we ask that it be changed.
The authors used the Bourin nomenclature for cell isolation although they do not test for positivity for CD74 (should be included in the text) but the adipose stromal vasculoskeletal fraction (SVF) is actually made up of double positive cells for CD34 and CD90 (this should also be included).
In addition to cell proliferation, it would also be useful to test viability using the MTT test or Calcein test.
The manuscript is very basic including cell proliferation, cytometry and differentiation.
It would be useful to implement it with molecular biology studies that usually confirm these data and by in vivo studies. I realize that it is currently not possible to carry out an in vivo study but the authors should include molecular biology studies to give a complete phenotype to these cells and above all to differentiation.
Reviewer 3 Report
Comments and Suggestions for Authors
The authors have described an automated isolation system to isolate and culture SVF components from the adipose tissue. While this automated system is intriguing, the authors did not fully explain or compare how this system works in comparison with current manual cell isolation systems. Major comments are listed below.
1. The doi and journal name for references are not right.
2. Since the main novelty of the manuscript is the automated isolation system for MSCs from the adipose tissue, it is recommended that Figure 1 and corresponding text needs to be moved to the Results section.
3. There is a general lack of control included in the study from Figure 3 to Figure 7.
4. The novelty of this automated cell isolation system does not look significantly improved from current manual isolation systems for SVF. The authors may modify their description of the system to make it more clear to readers what the improvement is.
5. Most cells in the SVF are CD45+ immune cells. The authors should perform flow cytometry analysis after 7 days in culture before the tri-lineage differentiation.
Comments on the Quality of English Languagena
Author Response
please see attached comments

Round 2
Reviewer 1 Report
Comments and Suggestions for Authors
The authors did not respond to many of the comments and suggestions raised. This result in the lack of substantial changes in the manuscript. Among the major issues raised that were not adequately answered by the authors were the following:
"The authors must show the nature of 3-D structures spheroids by staining with adipocyte and chondrocyte specific markers."
Perform a quantitative real time PCR to check the expression levels of adipocyte, chondrocyte and osteoblast specific markers to show that each SVF isolated samples is able to multilineage differentiate. Represents with dots-box plot".
"Figure 7 Use a table to show the data of SVF composition from each human samples. Moreover, represents the data in the histogram as dots-box plot to see the data from each individual."
I think that the manuscript cannot be accepted in the present form.
Comments on the Quality of English Language
Minor revision.
Reviewer 2 Report
Comments and Suggestions for Authors
The authors have partially addressed my previous concerns.
Please perform:
- test viability... It's useful also in colture test
- molecular biological studies (see my previous report)
Author Response
please see round 3 response
Reviewer 3 Report
Comments and Suggestions for Authors
Thanks to the authors for explaining the automated system in the response. While the authors try to address the advantage of the system in the discussion, the main improvement of the isolation protocol lies only in the automated washing and centrifugation steps. I agree with the authors that sterility is important in scaling up the culture system. However, due to the lack of controls throughout the paper, I am not convinced that this will lead to significant enough advance in the field. I also maintain concern in the lack of novelty in the manuscript.
Comments on the Quality of English Languagena
Round 3
Reviewer 2 Report
Comments and Suggestions for Authors
The authors have partially addressed my previous concerns.
Please perform:
- test viability... It's useful also in colture test
- molecular biological studies (see my previous report)
Author Response
We appreciate your time, energy and efforts in reviewing the manuscript. We did not see any new specific suggestions.
Reviewer 3 Report
Comments and Suggestions for Authors
na
Author Response
We appreciate your valuable feedback and have taken your comments seriously in our revised manuscript. We understand that the viability test is useful also in culture and it could demonstrate how these cells function under various conditions, including their ability to survive and thrive. The viability tests can provide additional insights into the cells and their activities, which are valuable for understanding the potential applications of these cells in the medical field and it was measure for the fresh SVF. This article aims to emphasize the characteristics of the cell sample in the final product derived from the automatic device. The final stromal vascular fraction (SVF), intended for human injection, pertains to cells that have not undergone cell culture processes. Any molecular biological work conducted on cell cultures does not characterize the product we aim to define.
We understand your request for viability testing, but unfortunately, due to the unavailability of specific cells and laboratory constraints, conducting additional biological experiments, including culture tests and viability assessments, is not feasible for this pilot work.
Regarding molecular biology studies and in vivo experiments, we acknowledge the importance of these aspects in providing a comprehensive understanding of cell behavior. However, due to the current limitations and constraints noted above, conducting further experiments of this nature is beyond the scope of this study. We will, however, acknowledge this limitation in the manuscript.
Thank you for your understanding and cooperation. If there are any additional specific points you would like us to address, please let us know.
Round 4
Reviewer 2 Report
Comments and Suggestions for Authors
The authors not performed novelty in this paper.